# Neck pain associated with smartphone usage among university students

**Mikhled Falah Maayah**[1,2]*, **Zakariya H. Nawasreh**[1], **Riziq Allah M. Gaowgzeh**[3], **Ziyad Neamatallah**[3], **Saad S. Alfawaz**[3], **Umar M. Alabasi**[3]

**1** Faculty of Applied Medical Sciences, Department of Rehabilitation Sciences, Division of physical therapy, Jordan University of Science and Technology, Irbid, Jordan, **2** Faculty of Medical Rehabilitation Sciences, Department of Occupational Therapy, King Abdulaziz University, Jeddah, Saudi Arabia, **3** Faculty of Medical Rehabilitation Sciences, Department of Physical Therapy, King Abdulaziz University, Jeddah, Saudi Arabia

* mikhledm@just.edu.jo

## Abstract

### Objective

Neck and shoulder pain has been linked to prolonged periods of flexed neck posture. However, the influences of factors related to individuals' characteristics and the time duration and position of using smartphones on the severity and duration of neck and shoulder pain among university students are not well studied. The aim of this study was to identify factors related to individual demographics, the history of neck pain, and the time duration and positions of using the smartphone that could be associated with neck pain severity and duration and to determine the influence of these factors on neck pain severity and duration among university students.

### Subjects and methods

A cross-sectional study was conducted on students from King Abdulaziz University in Jeddah, Saudi Arabia, using a self-administered online questionnaire. Data was collected between March 10th, 2020, and October 18th, 2020, with 867 questionnaires filled out using Google Forms as a web-based questionnaire. Questionnaires were distributed to students by posting them in their batch groups on Facebook, an online social media and social networking service. Students from five healthcare faculties were included: the faculties of medicine, dentistry, pharmacy, nursing, and medical rehabilitation sciences.

### Results

Students' gender, time spent on using their phones, time spent on devices for studying, and having a history of neck or shoulder pain were significant predictors of neck pain duration in the univariate model (p≤0.018). In the multivariate model, both having a history of neck or shoulder pain (95%CI: -2.357 to -1.268, p<0.001) and the hand-side used for writing (95% CI: 0.254–0.512, p<0.001) were significant predictors of neck pain severity, and they both explained 8.4% of its variance. A previous history of neck and shoulder pain, as well as time spent studying on devices, were predictors of the duration of neck pain. According to a study by researchers at Cardiff University, the hand side used for writing on smart devices was

**Data Availability Statement:** The data underlying the results presented in the study are available at https://figshare.com/s/ac23748dd94494111712.

**Funding:** The authors received no specific funding for this work.

**Competing interests:** The authors have declared that no competing interests exist.

also a good predictor of the severity of neck pain. A history of neck or shoulder pain (95% CI: 0.567–0.738, p = <0.001) and the number of hours spent on the device for studying (95% CI: 0.254–0.512, p<0.001) were significant predictors of neck and shoulder pain duration, and they both explained 8.4% of its variance. While having a history of neck or shoulder pain (95% CI: 0.639–0.748, p<0.001) and the hand-side used for writing (95% CI: -1.18 - -0.081, p = 0.025) were significant predictors of neck and shoulder pain severity, they explained 11.3% of its variance.

## Conclusions

The results of this study may be utilized to pinpoint smartphone usage factors associated with neck and shoulder pain severity and duration. Further, the findings of this study might help to develop preventive strategies to lower the impacts of these factors on the development of neck and shoulder pain severity and duration among university students.

## Introduction

Smartphones are now the most common portable electronic device used in the world [1–4]. In the current digital world, the use of smartphone technologies and applications has increased rapidly among university students [5–7]. This could be attributed to the fact that smartphone provide multifunctional ability to implement several functions such as information, communication, online learning, and enjoyment [8]. In recent studies on the prevalence of smartphone use, university students scored higher than any other age groups. As a result, a high significant proportion of musculoskeletal disorders, particularly in the neck, have been documented among university students [4,5]. In 2019, the age-standardized prevalence rate of neck pain was 27.0 per 1000 people, making it one of the most frequent musculoskeletal illnesses [9].

Saudi Arabia has become a more prosperous and educated community in recent years [10]. Saudis are now among the world's most devoted smartphone users. Many Saudis associate smartphone use with a more modern way of life [10]. Over 1.7 million students are currently enrolled in Saudi colleges and universities [11]. According to studies from two different Saudi universities, almost all university students owned a smartphone [12], and in 2016, 95% of university students used smartphones to access social media networks [13]. Following the coronavirus pandemic, most academic institutions worldwide, including Saudi Arabia, adapted either the online or the hybrid (in person and online) teaching methods for academic courses, which in turn increased the students' demand for using the smart devices including the smartphones [14,15].

Neck pain has become a prominent health problem in recent decades, with significant socioeconomic consequences for individuals, families, communities, and the healthcare system [2,16]. It was reported that between 8.2% and 90% of musculoskeletal pain in different body parts was due to the use of smartphones [17]. In Canada, 84% of students who use smartphones experienced musculoskeletal pain, with the neck being the most reported body part to experience pain [18]. In Saudi Arabia, neck and upper extremity pain were reported to account for 71% of patients with musculoskeletal pain, which is considered relatively higher than those reported for other countries, including Malaysia, Thailand, Nigeria, and Sudan [19].

Having musculoskeletal pain, including pain in the neck and upper extremities, may affect individuals' work productivity, functional performance, and quality of life. Further, it may

increase the demand for medical services and cause a substantial burden due to the cost of medical treatment. For students, experiencing musculoskeletal pain may impact their educational achievement and truancy from classes [20].

While using smart phones, there is a lack of neck and upper-limb support, combined with repeated finger movements for testing, which can result in a high static muscle load, especially when using only one hand [21–24]. Furthermore, forward head position while using smart phones has been identified as a risk factor for musculoskeletal pain, increasing tension on cervical structures and potentially initiating degeneration and tearing of the neck structures. A history of previous neck and shoulder pain may also increase the likelihood of developing recurrent neck pain and the severity of the condition. While there is an association between using an electronic devices (i.e., a smartphone) and musculoskeletal symptoms including neck pain among collegiate students, there is limited evidence regarding the influence of the students' demographics (sex and age), having an episode of previous neck pain, and various aspects of smartphone exposures including the smartphone time duration spent on using the phone, the number of typed massages, number of hands used to hold the device, the head positions of the smartphone, and associated musculoskeletal neck and shoulder pain duration and severity among university students. The aim of this study was to identify factors related to individual demographics, the history of neck pain, and the time duration and positions of using the smartphone that could be associated with neck pain severity and duration and to determine the influence of these factors on neck pain severity and duration among university students.

## Subjects and methods

### Data collection and study design

A cross-sectional study was conducted on students from King Abdulaziz University in Jeddah, Saudi Arabia, using a self-administered online questionnaire. Data was collected between March 10th, 2020, and October 18th, 2020, with 867 questionnaires filled out using Google Forms as a web-based questionnaire. Questionnaires were distributed to students by posting them in their batch groups on Facebook, an online social media and social networking service. Students from five healthcare faculties were included: the faculties of medicine, dentistry, pharmacy, nursing, and medical rehabilitation sciences. Furthermore, participants indicated whether they had experienced neck and shoulder pain related to the usage of a smart phone, and they rated the severity of the pain using the NRS-11 and the duration of the experienced pain. Univariate and multivariate linear regression models were used to identify the best fit model for neck and shoulder pain.

### Questionnaire design

First, general demographics, including age, gender, and faculty, were studied. Then the general conditions of mobile phone use, including handedness, frequency of mobile phone use, duration, and position during use, were studied. Students' experience of neck and shoulder pain associated with mobile phone use, including the severity of the pain using the NRS-11, was evaluated. Students were asked to rate their pain on a scale from 0 to 10, where zero represents "no pain at all" and 10 represents "the worst pain they have ever experienced," using whole numbers [25]. The English version of the survey was translated into Arabic using back translation for the Arabic sample (Kristine Bandgaard 2019). Finally, the measures students take to relieve their pain, including changing their position, decreasing their mobile phone use, using analgesics, and seeking medical care, were assessed.

## Inclusion and exclusion criteria

The study sample included students age 18 or older from the aforementioned five health care faculties, regardless of their age, gender, or handedness. Students who have been regularly using smartphones were included in this study. Students were excluded from the study if they had any of the following chronic medical conditions: musculoskeletal disorders or a prior surgery in the neck, shoulder, or upper limb.

## Ethical approval

The study protocal was approved by the Institutional Review Board (IRB) (date of approval: March 2, 2020) of the faculty of medical rehabilitation sciences at King Abdulaziz University. A written informed consent was obtained on the first page of the study's questionnaire, which was written in Arabic and English (the official language in Saudi Arabia is Arabic). The consent form explained the aims of the study and emphasized the confidentiality of the filled-out information. Participants were able to withdraw from the questionnaire at any point. No identifying information was obtained through the questionnaire, and all collected data were solely used for statistical analysis.

## Statistical analysis

The data was analyzed and described using percentages and frequencies. Statistical analysis was analyzed by using Statistical Package for Social Sciences Version 22 (SPSS) for Windows (SPSS Inc., Chicago, IL, USA). Univariate linear regression model was used to evaluate whether variables (independent variables) related to the conditions of mobile phone use (handedness, frequency of mobile phone use, duration, number of texts they send per day, and position during which they use the mobile phone, duration of using the phone for studying) and having a history of neck or shoulder pain can predict either the neck and shoulder pain duration in hours and the neck and shoulder pain severity (dependent variable) among collegiate students. Significant univariate predictors were then entered into the multivariate linear regression model with stepwise selection used to identify the best fit model for neck and shoulder pain severity and duration among university students (R2: R square, OR: odds ratios, and 95% confidence interval (95% CI)). P <0.05 was used to determine statistical significance. G*Power software v3.1.0 (Universität Düsseldorf, Düsseldorf, Germany) was used to determine the sample size for this study. The priori power analysis suggested that the minimum calculated sample size was 643 participants with a confidence level of 99%, a margin of error of 5%, and a response distribution of 50%. However, considering potential attrition and to improve the accuracy of the predictor model, an additional 224 partpicipants were included.

## Results

### Demographics

Overall, the current study involved 867 (86.7%) university students. The mean age of study participants was 21±2.95. There were 501 males (57.8%) and 366 females among the participants (42.2%) (Fig 1).

Most of the participating university students were from the faculty of medicine (24.80%), followed by the faculty of pharmacy, (21.22%), and the faculty of dentistry (19.84%), whereas the lowest students were from the faculty of rehabilitation (17.30%), followed by the faculty of nursing (16.84%), as shown in Fig 2.

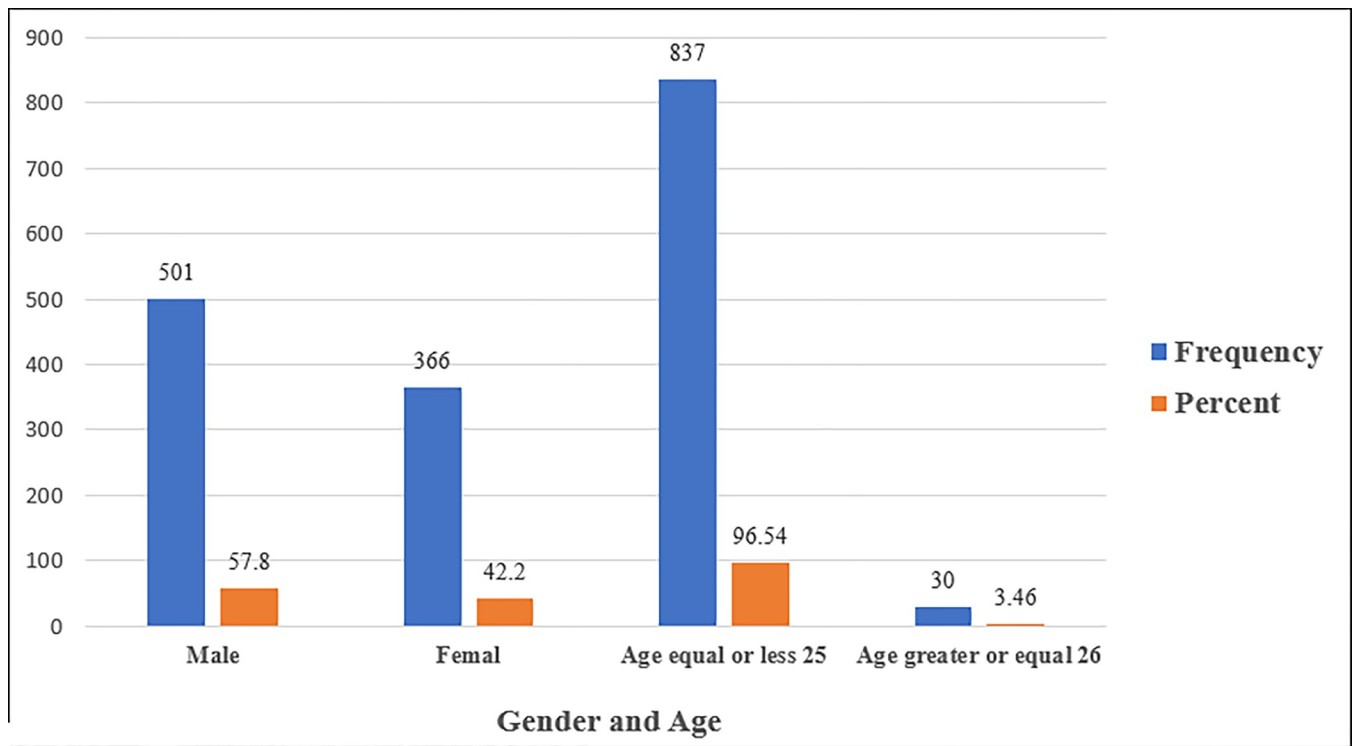

**Fig 1. Shows the frequency and percentage measures of neck pain by gender and age among university students.** Overall, the current study involved 867 (86.7%) university students. There were 501 males (57.8%) and 366 females among the participants (42.2%).

### Pattern of mobile phone use

In Table 1, among all the participants, 91.5% were right-handed dominant, and only 8.5% were left-handed dominant. The average daily frequency of smartphone use in days/week was 798 (92.0%) for all days, but the lowest was 20 (2.3%) for 1–2 days. The overall average of hours per day spent on smartphones revealed a maximum percentage of 465 (53.63%) equal to or greater than 4 hours and the lowest percentage of 402 (46.37%) less than 4 hours. When considering the purpose of using smartphones, it was found that the highest percentage, 37.14%, was for study purposes, while the lowest percentage, 1.15%, was for studying and playing games.

For test massages sent per day, the highest percentage was for those who sent 22 or more test massages: 317 (36.56%), followed by 1–5 messages with 260 (29.99%), and the lowest percentages of 12–15 with 32 (3.69%). The percentage of mobile phone use in a single or both hands showed that the single hand was higher (63.44%) than using it with both hands (36.56%). The most common position when using a cell phone was sitting (61.48%), then supine lying (33.68%), and finally standing and walking (both 4.84%) (Table 1).

In Table 1, there were 72.55% of the students who reported having neck or shoulder pain. In addition, the highest site of pain was in the neck at 48.44%, followed by the right shoulder at 16.03%, and then the left shoulder at 6.57%. As for the pain duration, the highest percentage was found to be 233 (26.87%) for one hour, while the lowest was 21 (2.42%) for five hours (Fig 3). Also, the most common time of pain was 279 (32.18%) at night, while the lowest percentage was 57 (6.57%) in the afternoon (Fig 4). The results of the current study showed that 292 (33.68%) had a pain intensity of less than 4/10 and 322 (37.2%) had a pain intensity higher than 4/10, whereas 253 (29.18%) reported no pain at all (Fig 5).

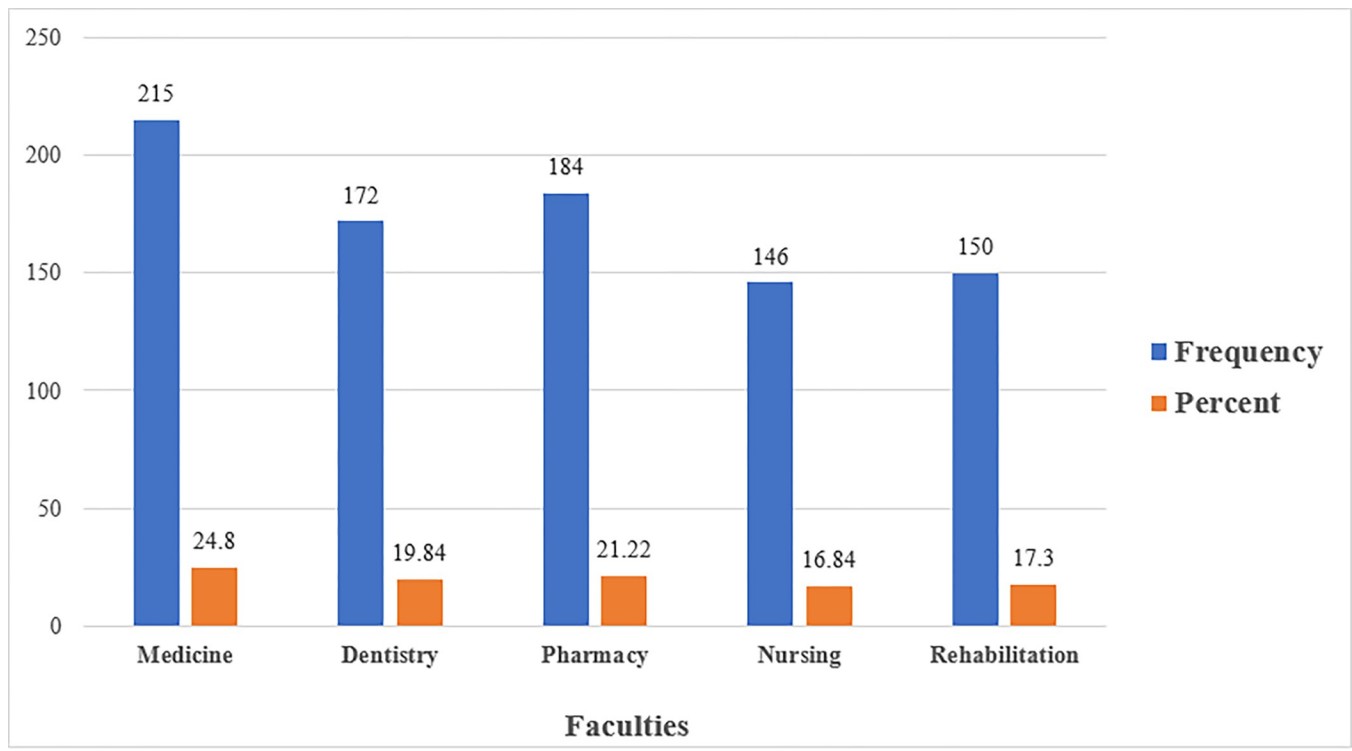

**Fig 2. Shows the frequency and percentage measures of neck pain by faculties among university students.** According to the figure, the majority of participating university students were from the faculty of medicine (24.80%), followed by the faculty of pharmacy (21.22%), and the faculty of dentistry (19.84%), with the lowest students coming from the faculty of rehabilitation (17.30%) and nursing (16.84%).

In Fig 6, the frequency and percentages in the results of the study revealed that the highest pain frequency in days per week was seen at 51 (5.88), followed by four days at 55 (6.33), and five days at 36 (4.15). In contrast, the lowest pain frequency per week is on day 198 (22.84), followed by 2 days 123 (14.19), and 3 days 121 (13.96).

In Table 2, the percentage of students who used pain-reducing analgesia was highest for No 703 (81.08%), and lowest for Yes 164 (18.92%). The percentage of students with varying degrees of pain severity was 796 (91.8%), "I did not seek medical attention for myself," but 56 (6.5%) Yes, I've visited the clinic, followed by 15 (1.7%) who visited the emergency department in clinics (Table 2). However, the number of students who changed position the most frequently after experiencing pain while using the mobile phone showed the percentage of "yes" at 470 (54.2%) and the "no" showed 255 (29.4%) for students who did not change their position (Table 2).

In Fig 7, the findings of the current study are shown to be that 292 (33.7%) of the participants had mild pain, 233 (26.9%) had moderate pain, and 89 (10.3%) had severe pain.

Students gender, average time spent using the phone, time spent on the device for studying, and having a history of neck and shoulder pain were significant predictors of neck and shoulder pain duration in the univariate model (p≤0.018, Table 3).

In the multivariate model, only having a history of neck or shoulder pain (95% CI: 0.567–0.738, p≤0.001) and the number of hours spent on the device for studying (95% CI: 0.254–0.512, p<0.001) were significant predictors of neck and shoulder pain duration, and they both explained 8.4% of its variance (Table 4).

In the univariate model, all hand sides used for writing text messages, time spent using a mobile device, and having a history of neck or shoulder pain were significant predictors of

**Table 1. Frequencies and percentages measures for mobile phone use.**

| Variable | Categories | Frequency (N) | Percent (%) |
|---|---|---|---|
| Handedness | Left-Handed | 74 | 8.54 |
| | Right-Handed | 793 | 91.46 |
| The average frequency of mobile phone use (days per week) | 1–2 Days | 20 | 2.31 |
| | 3–4 days | 49 | 5.65 |
| | All days | 798 | 92.04 |
| The average time spent using a mobile phone (Hour) | 1–2 | 71 | 8.2 |
| | 3–4 | 201 | 23.2 |
| | 4–5 | 130 | 15.0 |
| | 6–7 | 229 | 26.4 |
| | 8–9 | 60 | 6.9 |
| | 10–12 | 92 | 10.6 |
| | ≥13 | 84 | 9.7 |
| Purpose of use | Playing Games | 12 | **1.39** |
| | Social Media | 177 | 20.42 |
| | Studying | 322 | 37.14 |
| | Studying and Playing Games | 10 | 1.15 |
| | Studying and social media | 253 | 29.18 |
| | Studying and Watching Videos | 46 | **5.31** |
| | Texting | 14 | **1.61** |
| | Watching Videos | 14 | **1.61** |
| | Working | 19 | 2.19 |
| Hours spent on the study tool (Hour) | 0 | 8 | 0.92 |
| | 1–2 | 661 | 76.24 |
| | 3–4 | 132 | 15.22 |
| | 5–6 | 45 | 5.19 |
| | ≥7 | 21 | 2.43 |
| Text messages sent per day | 1–5 | 260 | 29. 99 |
| | 6–10 | 157 | 18.11 |
| | 12–15 | 32 | 3.69 |
| | 16–20 | 101 | 11.65 |
| | ≥22 | 317 | 36.56 |
| Single Handed or both handed | One hand | 550 | 63.44 |
| | Both hands | 317 | 36.56 |
| The position adopted while in use | Sitting position | 533 | 61.48 |
| | Standing position | 21 | 2.42 |
| | Walking position | 21 | 2.42 |
| | Supine position (Lying down) | 292 | 33.68 |
| Experience neck or shoulder pain before | No | 238 | 27.45 |
| | Yes | 629 | 72.55 |
| Pain site | Neck | 420 | 48.44 |
| | Right shoulder | 139 | 16.03 |
| | Left shoulder | 57 | 6.57 |
| | I do not have pain | 251 | 28.95 |

neck and shoulder pain severity (p = 0.018) (Table 5). However, in the multivariate model, both having a history of neck or shoulder pain (95% CI: 0.639–0.748, p<0.001) and the hand-side used for writing (95%CI: -1.18 - -0.081, p = 0.025) were significant predictors of neck and shoulder pain severity, and they both explained 11.3% of its variance. (Table 6).

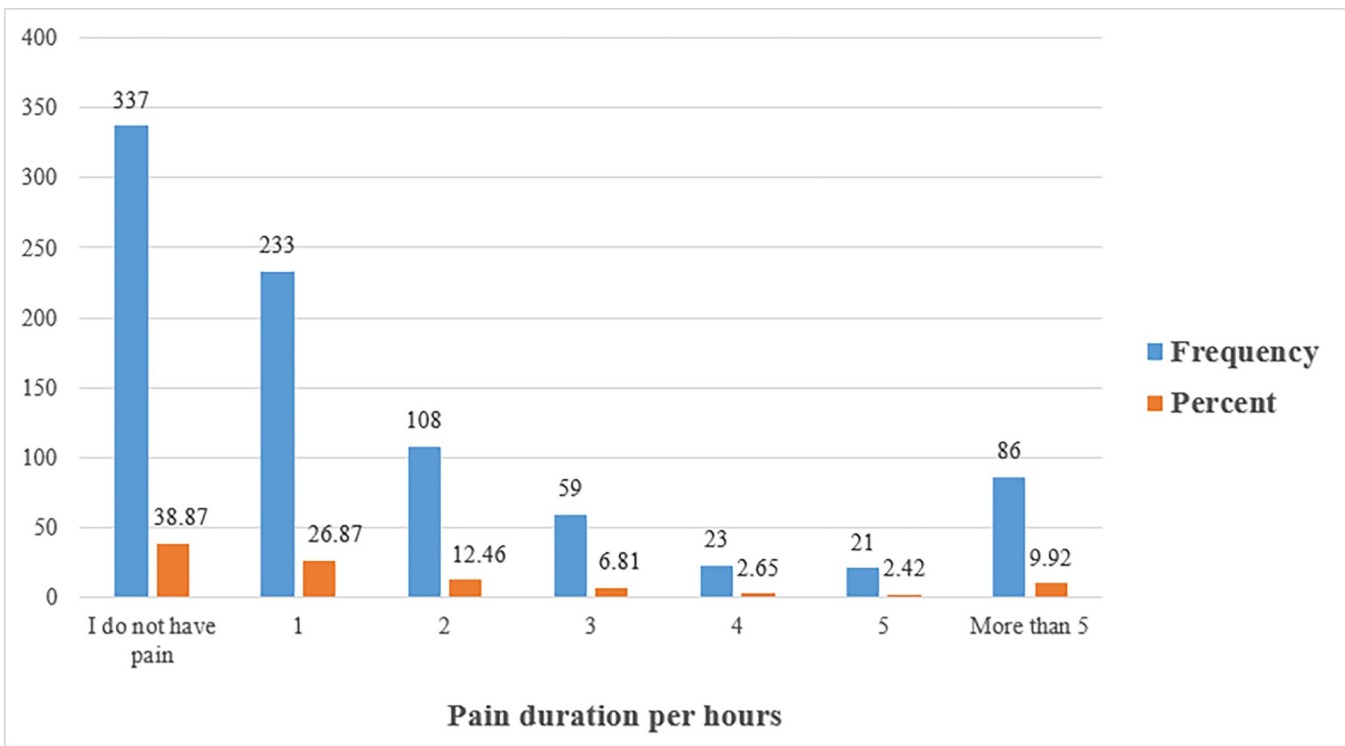

**Fig 3. Shows the frequency and percentage measures of neck pain by pain duration per hours among university students.** As for the pain duration in this figure, the highest percentage was found to be 233 (26.87%) for one hour, while the lowest was 21 (2.42%) for five hours.

## Discussion

The purpose of the current study was to identify factors related to individual demographic usage of the smart phone and smartphone exposure that are associating with neck and shoulder pain duration and intensity, and to determine the influence of these factors on neck and shoulder pain duration and intensity among university students. According to the findings of this study, neck and shoulder pain duration is associated with all of student's sex, time spent on phone in studying, and having a history of neck and shoulder pain. However, the multivariate model revealed that only having a history of neck and shoulder pain, as well as the number of hours spent studying on a device, were significant predictors of the duration of neck pain. The hand-side use for writing on smartphone, time spent on mobile devices, and having a history of neck and shoulder pain were individual predictors of the severity of neck pain. Yet, the multivariate model revealed that only having a history of neck and shoulder pain and the hand side used for writing were significant predictors of the severity of the neck and shoulder pain. The findings of this study may help identify factors associated with neck and shoulder pain duration and severity among academic students. Further, they might help to develop preventive strategies to lower their impacts of these factors on the development of neck pain among university students.

The percentages of neck and shoulder pain in the present study revealed that 72.6% of participants used smartphones. This rate was similar to a previous study that reported that 71.2% of subjects suffered from cervical pain, which was the most common symptom [26]. In the current study, university students who use smartphones for studying recorded a significantly higher percentage than those who use smartphones for other purposes. There is a major difference in gender levels of pain severity among university students. There were 49 (5.7%) females

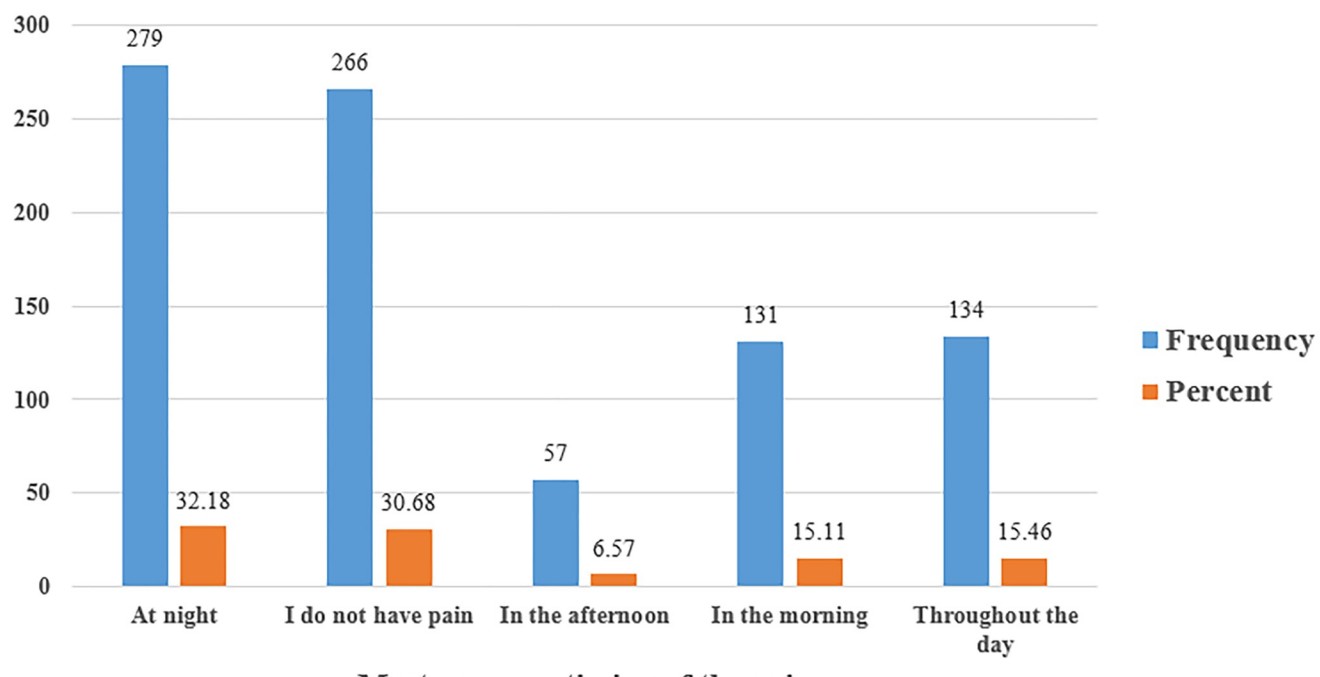

**Fig 4. Shows the frequency and percentage measures of neck pain by the most common timing of the pain among university students.** For the pain duration shown in this figure, the most common time of pain was 279 (32.18%) at night, while the lowest percentage was 57 (6.57%) in the afternoon.

with severe pain, while 40 (4.61%) males had severe pain, which can be referred to as females who use more mobile phones than males.

The findings from the current study showed that the most common timing of pain was at night (32.2%), while it was less frequent in the afternoon (6.6%). In 2014–2015, a survey of 2,367 Saudi University students was conducted [2]. In this survey, the percentages among university students showed that 27.2% reported using their phones for more than 8 hours per day, while 75% reported using them for less than four 4 hours per day [3]. In contrast, Amal et al. [27] conducted a recent study in Saudi Arabia and revealed that 45% of participants were using smartphones, with 35.1% spending 6–9 hours on average. Furthermore, 40.5% used mobile phones with one hand with a slight neck tilt below the horizontal line; 59.1% reported neck or shoulder pain while using devices; and only 2.7% of those experiencing pain used pain relief medications. In the current study, 63.4% of university students used smartphones with one hand, while 23.5% of those with neck pain used medications to relieve their pain. Instead of analgesics, 10.6% of participants used massage, rest, ice, and heat modalities. According to the findings of the current study, university students spend more hours studying (37.14%), which is the highest percentage of the purpose of using mobile tools, while 17.19% spend over 6 hours studying, which is the most common factor that is associated with the neck pain.

The findings of this study revealed that a student's sex, time spent on a smartphone while studying, and having a history of neck and shoulder pain were associated with the neck and shoulder pain, with having a previous history of pain being the strongest predictor, followed by the number of hours spent on devices while studying, and they explained 4.8% and 3.9% of the variance of the pain duration, respectively. Individuals with a previous history of neck or shoulder pain were more likely to have a longer pain duration than those with no previous episodes of pain, **and** those who spent more time studying on smart devices were more likely to

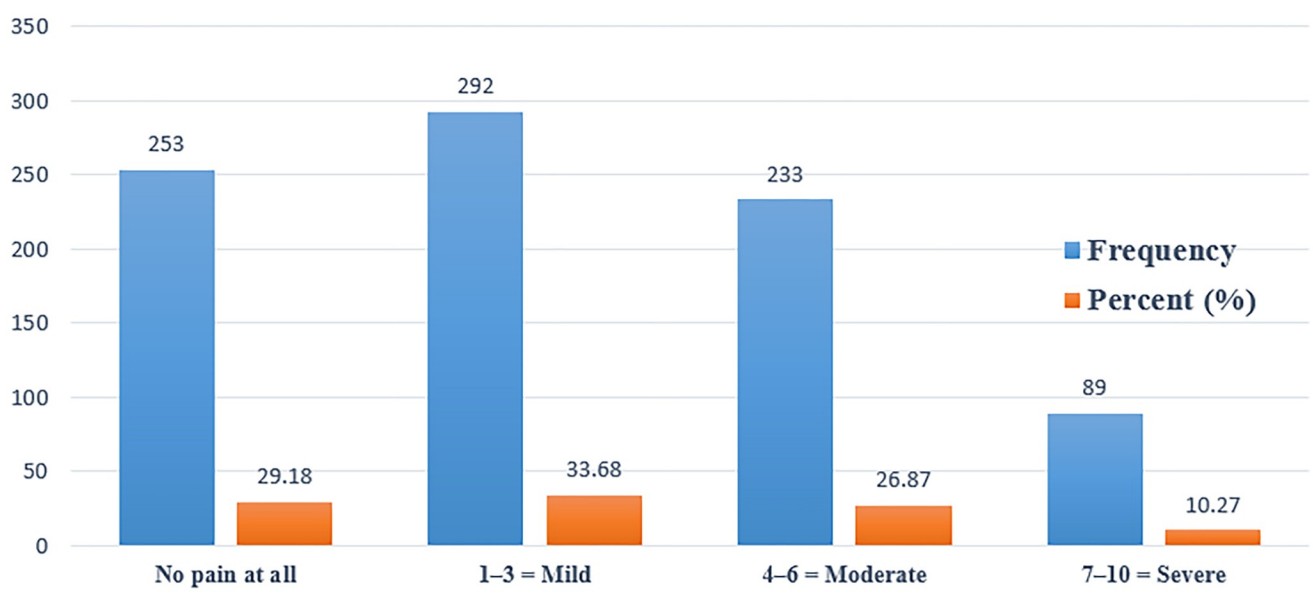

**Fig 5. Shows the frequency and percentage measures of neck pain by pain intensity (on a scale of 1–10) among university students.** In this figure, the results of the current study show that 292 (33.68%) had a pain intensity of less than 4/10 and 322 (37.2%) had a pain intensity higher than 4/10, whereas 253 (29.18%) reported no pain at all.

have a longer neck pain duration. While there are no previous studies that reported on the influence of having a history of neck pain due to the usage of smart phone and the neck pain duration among university students, it was reported that the development of chronic musculoskeletal pain was associated with history of low back pain and high initial pain intensity [28]. It could be that the usage of smart phone has just recently increased in number, which could explain the lack of studies that have reported on the influence of having a history of neck symptoms on develop further episodes of neck and shoulder pain among collegiate students who use their smartphones for education purposes. In a study by Al-Hadidi et al, [29] reported that the duration of mobile phone use had a positive correlation with the duration and severity of neck pain among university students. However, in a meta-analysis that included 9 studies revealed inconclusive findings about the time duration of using smartphone and musculoskeletal pain, yet 6 of the studies included in the meta-analysis indicated an association between the time spent on the smartphone and musculoskeletal pain [29]. The findings of the current study alongside the previous studies do not specify the exact cut-off value of time that associate with developing musculoskeletal pain. Therefore, further studies may be needed to determine the cut-off point for the time of using the smartphone that could lead to the development of musculoskeletal pain to provide recommendation for university students.

The findings of this study revealed that the hand side commonly used for writing on smartphones, time spent on mobile devices, and having a history of neck and shoulder pain were individual predictors of neck and shoulder pain severity. With the time spent using a phone being the strongest predictor, followed by having a previous history of neck or shoulder pain, they explained 5% and 4.8% of the variance of neck pain severity, respectively. However, the multivariate model revealed that only having a history of neck and shoulder pain and the hand

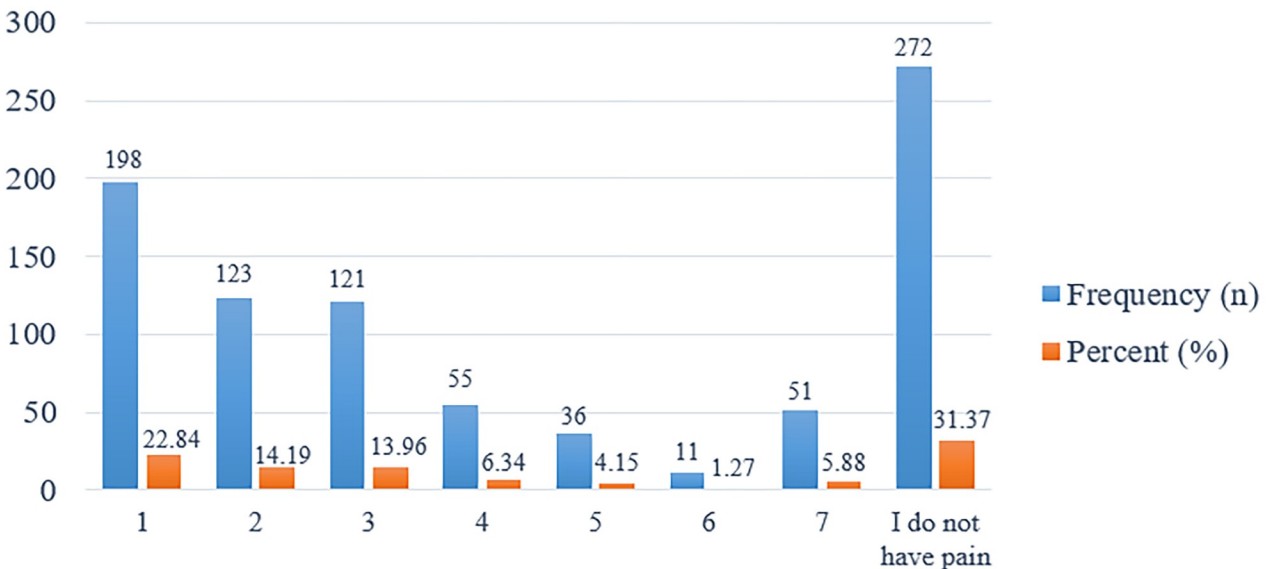

**Fig 6. Shows the frequency and percentage measures of neck pain by pain frequency (in days per week) among university students.** The figure shows the frequency and percentages in the results of the study, which revealed that the highest pain frequency in days per week was seen at 51 (5.88), followed by four days at 55 (6.33), and five days at 36 (4.15). In contrast, the lowest pain frequency per week is on day 198 (22.84), followed by 2 days 123 (14.19), and 3 days 121 (13.96).

side used for writing on a smartphone were significant predictors of the neck and shoulder pain severity and they explained 11.3% of the variance. Individuals with a history of neck or shoulder pain tend to be at higher risk to have severe neck pain than those with no history of neck or shoulder pain, while those who type with their left hand tend to be less likely to have severe neck pain than those who type with their right hand. This indicate that students with previous episode of neck pain are more likely to have a severe neck pain. It can be advocated that having a previous episode of neck or shoulder pain may have caused biomechanical and flexibility changes, proprioceptive deficits, range of motion limitation, muscle spasm or weakness, and degenerative changes in the neck and shoulders that, in turn, may lead to sustain a more severe neck and shoulder pain due to the usage of smartphones. Further, there might be negative impacts of previous episode of neck pain on students, especially in case that students may have not fully recovered or did not receive sufficient treatment to ensure resolving symptoms and impairment of previous episode. Students who type with their right hand are at higher risk of developing severe neck and shoulder pain compared to those who type with their left hand. This may have resulted from the high volume of repetitive movements performed by the right hand. This could be since 91.46% of the sample included in this study are right hand dominant. Therefore, adding long periods of typing on smartphones to their right-hand activity requires placing their upper extremity up while holding their phones so the smartphone is within the range of their vision, which places a high demand on the neck and shoulder muscles. This, in turn, may cause muscle fatigue, and changes in the balance of muscle strength and length that affect the cervical spine and scapula movements and alter their biomechanical loads could result in severe neck and shoulder pain. Further, it could be that those

**Table 2. Shows the frequency and percentages of people who use analgesics and seek medical care for neck pain.**

| Variable | Categories | Frequency (N) | Percent (%) |
|---|---|---|---|
| **Analgesia used to decrease pain** | No | 703 | 81.08 |
| | Yes | 164 | 18.92 |
| **Types of Analgesic agents used** | I did not take any medications | 652 | 75.2 |
| | Alternatives to Analgesics: Heat | 10 | 1.2 |
| | Alternatives to Analgesics: Ice | 2 | 0.2 |
| | Alternatives to Analgesics: Massage | 28 | 3.2 |
| | Alternatives to Analgesics: Relaxation techniques | 4 | 0.5 |
| | Alternatives to Analgesics: Rest | 38 | 4.4 |
| | Neurological Analgesia (gabapentin, amitriptyline etc) | 3 | 0.3 |
| | Nonsteroidal Anti-inflammatory Drugs (ibuprofen, Panadol naproxen, and Aspirant) | 130 | 15.0 |
| **The frequency of the use of analgesics agents (In Day per week)** | 0 | 690 | 79.6 |
| | 1 | 100 | 11.5 |
| | 2 | 32 | 3.7 |
| | 3 | 31 | 3,6 |
| | 4 | 2 | 0.2 |
| | 5 | 5 | 0.6 |
| | 6 | 7 | 0.8 |
| **Seeking medical care to relieve pain** | No | 796 | 91.8 |
| | Yes, I've visited the Clinic | 56 | 6,5 |
| | Yes, I've visited the Emergency Department | 15 | 1.7 |
| **Did You Decrease the use of the mobile phone after experiencing pain** | I do not have pain | 143 | 16.5 |
| | No | 528 | 60.9 |
| | Yes | 196 | 22.61 |
| **Change the most frequent position while using a mobile phone after experiencing this pain** | I do not have pain | 142 | 16.4 |
| | No | 255 | 29.4 |
| | Yes | 470 | 54.2 |
| **Rating the pain intensity decreased by using a pain intensity scale of 10** | 1/10 | 457 | 53.9 |
| | 2/10 | 261 | 30.1 |
| | 3/10 | 139 | 15 |
| **Do you think that this pain was related to the use of your mobile phone?** | I do not have pain, and I Do not believe that this type of pain is related to mobile phones use. | 37 | 4.3 |
| | I do not have pain, but I believe that this type of pain is related to mobile phones use | 126 | 14.5 |
| | No | 212 | 24.5 |
| | Yes | 492 | 56.7 |

who use their right hand may have less resting time when added to the daily routine activities performed in the right hand. Therefore, students who are right-hand dominant might be advised to minimize the time of using their phone in right hand, and instead used their left hand for typing.

The current study's findings revealed that the most common handedness employed with a high percentage correlation between neck and pain intensity level was single-handed use, which had mild pain that reached 191 (0.3%). The sitting position had a greater percentage of 59 (0.111%), while the lowest percentage was 2 (0.095%) for (standing and walking positions. The duration of use of smartphone plays a key role in assessing the duration of pain in the neck and shoulders.

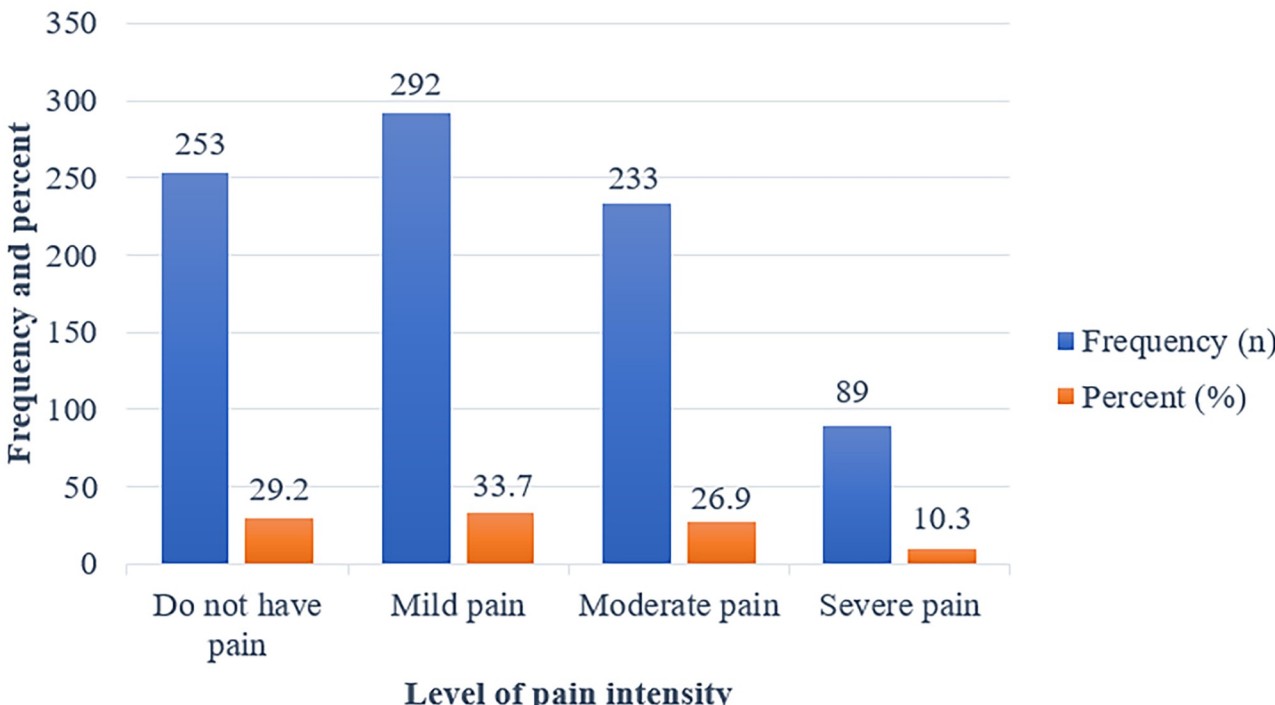

**Fig 7. Shows the frequency and percentage measures of neck pain by level of intensity among university students.** This figure shows the findings of the current study, which are that 292 (33.7%) of the participants have mild pain, 233 (26.7%) have moderate pain, and 89 (24.9%) have severe pain.

**Table 3. Univariate linear regression for predictor measures of neck and shoulder pain duration in hours among universities students.**

| | β coefficients | $R^2$ | 95%CI | P-value |
|---|---|---|---|---|
| **Gender (Male, Female*)** | **0.616** | **0.6%** | **0.105–1.128** | **0.018** |
| **Age (year)** | -0.045 | 0.1% | -0.137–0.047 | 0.339 |
| **Which hand do you use for writing? (Right*, left)** | -0.401 | 0.1% | -1.308–0.506 | 0.386 |
| **Average frequency for your use of your mobile device (in days per week)** | 0.104 | 1% | -0.109–0.318 | 0.338 |
| **Average time do you spend using your mobile device? (in hours per day)** | **0.085** | **1.0%** | **0.029–0.142** | **0.003** |
| **How many Hours do you spend on your device for studying? (hours)** | **0.399** | **3.9%** | **0.267–0.531** | **0.001** |
| **How many text messages do you send per day?** | 0.002 | 0.2% | -0.001–0.005 | 0.221 |
| **Do you hold your mobile in one hand or both hands when you use it (One hand, two hands*)** | 0.373 | 0.2% | -0.153–0.899 | 0.164 |
| **What is your most frequent position when you use your mobile device?** | -0.089 | 0.01% | -0.269–0.092 | 0.335 |
| **Did you experience neck or shoulder pain before? (Yes, No*)** | **-0.435** | **4.8%** | **0.567–0.738** | **0.001** |

Boldfaced **P values** indicate statistical significance; **B**: B coefficient, **CI**: Confidence interval; **SD**: Standard Deviation; * Reference group for each predictor.

**Table 4. Multivariate linear regression for predictor measures of neck and shoulder pain duration in hours among universities students.**

| | β coefficients | $R^2$ | 95%CI | P-value |
|---|---|---|---|---|
| **Did you experience neck or shoulder pain before (Yes*, No)** | -0.435 | 8.4% | 0.567–0.738 | <0.001 |
| **How many hours do you spend on your device for studying?** | 0.383 | | 0.254–0.512 | <0.001 |

**Table 5.** Univariate linear regression for predictor measures of neck and shoulder pain severity among universities students.

| | β coefficients | $R^2$ | 95%CI | P-value |
|---|---|---|---|---|
| **Gender (Male, Female**\***)** | 0.205 | 0.2% | -0.125–0.535 | 0.223 |
| **Age (year)** | 0.055 | 0.4% | -0.004–0.115 | 0.066 |
| **Which hand do you use for writing? (Right**\***, left)** | **-0.106** | **0.5%** | **-1.77- -0.013** | **0.045** |
| **Average frequency for your use of your mobile device (in days per week)** | 0.118 | 0.3% | -0.019–0.255 | 0.092 |
| **Average time (in hours per day) do you spend using your mobile device** | **0.038** | **5%** | **0.001–0.074** | **0.042** |
| **How many Hours do you spend on your device for studying?** | 0.035 | 0.1% | -0.051–0.122 | 0.425 |
| **How many text messages do you send per day?** | 0.001 | 0.1% | -0.001–0.003 | 0.281 |
| **Do you hold your mobile in one hand or both hands when you use it.** | -0.048 | 0.1% | -0.387–0.290 | 0.779 |
| **What is your most frequent position when you use your mobile device?** | -0.089 | 0.01% | -0.269–0.092 | 0.335 |
| **What is your most frequent position when you use your mobile device?** | -0.003 | 0% | -0.119–0.113 | 0.955 |
| **Did you experience neck or shoulder pain before (Yes, No**\***)** | **-0.369** | **4.8%** | **0.639–0.748** | **0.001** |

Boldfaced **P** values indicate statistical significance; **B**: B coefficient; **CI**: Confidence interval; **SD**: Standard Deviation; \* Reference group for each predictor.

The findings of the current study revealed that the student gender was an individual predictor of the neck and shoulder pain duration, with male students (n = 366) (42.2%) were more likely to experience longer neck and shoulder pain than female students due to the usage of smartphone devices. However, it is unknown how students' sex influences the neck and shoulder pain duration.

These findings of the current study may have clinical implications as they help identify factors related to individual demographics and smartphone usage associated with the duration and severity of neck and shoulder pain. The findings of this study suggest that students with a previous history of neck pain and who spend more time using smartphones are more likely to develop neck pain that lasts for a long time. Therefore, it is recommended that these students may adapt preventive strategies by minimize the time spent using their smartphones. This can be achieved by dividing the time needed for studying on a smart phone into small segments and considering taking frequent rest time to avoid developing neck pain. Further, the findings of this study indicate that students with a history of neck pain and who use one hand to hold the phone are more likely to develop severe neck pain. Therefore, it is recommended that they alternate using their phones between their right and left hands or to use both hands especially when they use their smartphones for long periods of time, such as when studying. To avoid long duration and severe neck and shoulder pain, students may be instructed to adopt an extended neck sitting position, supporting forearms, holding a mobile phone with both hands, and using both thumbs without sustaining a position for an extended period of time [21,22].

The findings of this study could also be used to develop prevention strategies and health promotion initiatives aimed at reducing neck and shoulder pain and addressing the risk of neck disorders associated with smartphone users. With smartphones are becoming increasingly relevant in all aspects of our lives, therefore more focus should be given to educating the public on the impact of long-time usage of smart phone. Further, individuals who use smart phone as part of their education or profession may be instructed on maintaining a healthy

**Table 6.** Multivariate linear regression for predictor measures of neck and shoulder pain severity in hours among university students.

| | β coefficients | $R^2$ | 95%CI | P-value |
|---|---|---|---|---|
| **Did you experience neck pain or shoulder before (Yes, No)** | -0.369 | | 0.639–0.748 | 0.001 |
| **Which hand do you use for writing (right**\***, left)** | -0.630 | 11.3% | -1.18 - -0.081 | 0.025 |

sitting positions, use both hands or alternate between hand while holding their phone, and ensue using the phone for short durations with rest interval to monitor the rising incidence of neck and upper extremity pain in our society. University students should understand and be aware of how to appropriately use smartphones and other technological tools. Furthermore, faculties should place more emphasis on raising awareness about the proper usage of smartphones and providing student with studying materials that require using smart phone for short time.

## Limitations of the study

Notwithstanding the appreciated data, there are some limitations to this study. Because the study used a cross-sectional design, it found significant associations among the evaluated independent variables. Furthermore, many of the independent variables evaluated are self-reported, which may cause biases. Because study data were gathered through convenience sampling, the study findings cannot be generalized to larger or similar populations. Also, in the current study, researchers did not consider the use of any other electronic devices, such as desktop computers or other study-related devices. It was possible to increase the number of participants; however, due to a lack of time, we were unable to think of more than those chosen for the study.

## Conclusions

Students' sex, time spent on the phone while studying, and having a history of neck and shoulder pain were individual predictors of neck and shoulder pain duration. However, only having a history of neck and shoulder pain and the number of hours spent studying on a device were predictors of the neck and shoulder pain duration in the multivariate model. While hand-side use for writing on a smartphone, time spent on smartphones, and having a history of neck and shoulder pain were individual predictors of the severity of neck and shoulder pain. Yet, only having a history of previous neck and shoulder pain and the hand-side used for writing were predictors of the severity of the neck and shoulder pain in the multivariate model. The findings of this study may help to pinpoint smartphone usage factors associated with neck and shoulder pain severity and duration. Further, the findings of this study might help to develop preventive strategies to lower the impacts of these factors on the development of neck and shoulder pain severity and duration among university students.

## Acknowledgments

The authors would like to thank the deans of research, the vice dean of scientific research, the heads of departments, and all of the university students who took the time to complete the survey questionnaire.

## Author Contributions

**Conceptualization:** Mikhled Falah Maayah, Zakariya H. Nawasreh, Ziyad Neamatallah, Umar M. Alabasi.

**Data curation:** Mikhled Falah Maayah, Riziq Allah M. Gaowgzeh, Ziyad Neamatallah, Saad S. Alfawaz.

**Formal analysis:** Zakariya H. Nawasreh, Riziq Allah M. Gaowgzeh.

**Investigation:** Mikhled Falah Maayah, Zakariya H. Nawasreh, Riziq Allah M. Gaowgzeh, Umar M. Alabasi.

**Methodology:** Mikhled Falah Maayah, Riziq Allah M. Gaowgzeh, Ziyad Neamatallah, Saad S. Alfawaz.

**Project administration:** Mikhled Falah Maayah.

**Resources:** Mikhled Falah Maayah.

**Supervision:** Mikhled Falah Maayah, Umar M. Alabasi.

**Validation:** Mikhled Falah Maayah, Umar M. Alabasi.

**Writing – original draft:** Mikhled Falah Maayah, Zakariya H. Nawasreh, Riziq Allah M. Gaowgzeh, Ziyad Neamatallah, Saad S. Alfawaz, Umar M. Alabasi.

**Writing – review & editing:** Mikhled Falah Maayah, Zakariya H. Nawasreh, Riziq Allah M. Gaowgzeh, Ziyad Neamatallah, Saad S. Alfawaz, Umar M. Alabasi.

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
