## [Decision Letter · Decision Letter 0]

25 Jan 2023

PONE-D-22-35345Neck pain associated with smartphone usage among university studentsPLOS ONE

Dear Dr. Maayah,

Thank you for submitting your manuscript to PLOS ONE. After careful consideration, we feel that it has merit but does not fully meet PLOS ONE’s publication criteria as it currently stands. Therefore, we invite you to submit a revised version of the manuscript that addresses the points raised during the review process.

I am very much thankful to the reviewers for their deep and thorough review.

Major revisions are required. Please check the reviewers’ comments.

We look forward to receiving your revised manuscript.

Kind regards,

Sıdıka Bulduk, Prof. Dr.

Academic Editor

PLOS ONE

Journal Requirements:

5. We note you have included a table to which you do not refer in the text of your manuscript. Please ensure that you refer to Table 5 and 6 in your text; if accepted, production will need this reference to link the reader to the Table.

Reviewers' comments:

Reviewer's Responses to Questions

**Comments to the Author**

1. Is the manuscript technically sound, and do the data support the conclusions?

Reviewer #1: Partly

Reviewer #2: Yes

2. Has the statistical analysis been performed appropriately and rigorously? 

Reviewer #1: I Don't Know

Reviewer #2: Yes

3. Have the authors made all data underlying the findings in their manuscript fully available?

Reviewer #1: No

Reviewer #2: Yes

4. Is the manuscript presented in an intelligible fashion and written in standard English?

Reviewer #1: Yes

Reviewer #2: Yes

5. Review Comments to the Author

Reviewer #1: For lines 91 and 92, please check out the numbers and the percentages.

For lines 96-98, please check out the numbers and the percentages.

Please add the word hour(s) to Table 1 for the variable - average time spent with mobile phones.

Please add the percentages for the variable purpose of use - playing games.

Page 12, for lines 238 and 239, please check out the percentages.

Page 13, for lines 249-250, it was written “(OR: -250 0.213, 95% CI: -2.357 to -1.268, p<0.001)” for Table 4. Really?

Page 13, for lines 258-260, repetition of the same paragraph which was written above.

It is stated that “(OR: -250 0.213, 95% CI: -2.357 to -1.268, p<0.001)” is written in Table 4 for lines 249-250 on page 13. Really?

According to me, in the discussion section, links/comparisons to other literature are poor. The discussion section should be rewritten.

Reviewer #2: Abstract Section

Mention the objectives correctly.Conclusion in the abstract section should not include the future scope. Mention clinical implication of the current study in the conclusion

Main Manuscript

Methods

1. Give statistics of how many questionnaires were rolled out and how many filled with Response rate calculation

2. Exclusion criteria to be mentioned

3. Process of development of questionnaire and how its validity was done to be mentioned

4. In inclusion criteria, give operational definition for duration of usage of smartphone.

Discussion to be written in detail with reason for all the factors to have an impact on neck pain

6. PLOS authors have the option to publish the peer review history of their article (what does this mean?). If published, this will include your full peer review and any attached files.

Reviewer #1: No

Reviewer #2: **Yes: **Prachita Walankar

---

## [Author Response · Author response to Decision Letter 0]

14 Mar 2023

PLOS ONE

March 9, 2023

Re: manuscript re-submission

Dear Editor,

We are resubmitting our manuscript, "Neck pain associated with smartphone use among university students," to be considered for publication as an original research article in the Healthcare Journal.

We thank you and the reviewers for taking the time to read our paper and providing valuable comments. It was your valuable and insightful comments that led to possible improvements in the current version. The authors have carefully considered the comments and tried our best to address every one of them. We are hopeful that the manuscript will careful revisions to meet your high standards. The authors welcome further constructive comments, if any.

Below, we provide the point-by-point responses. All modifications to the manuscript have been highlighted in red. Please refer to our submission page and line numbers to see what modifications we've made. The authors provided two copies of the manuscript: one with tracking changes and the other without. 

Thank you for your time and consideration of our work. If you have any questions, please contact us.

Sincerely,

Journal Requirements:

The followings were added to the ethical approval section. “A written informed consent was obtained on the first page of the study’s questionnaire, which was written in Arabic and English (the official language in Saudi Arabia is Arabic). For participants aged less than18 years, their parents/guardians also provided informed written consent.”

We are very sorry

The data underlying the results presented in the study have been submitted.

The data underlying the results presented in the study have been submitted.,

5. We note you have included a table to which you do not refer in the text of your manuscript. Please ensure that you refer to Table 5 and 6 in your text; if accepted, production will need this reference to link the reader to the Table.

Tables 5 and 6 were references in the result section. Has been corrected to Table 5 instead of Table 3 in the clean copy as per your request, as follows: 

“In the univariate model, all hand sides used for writing text messages, time spent using a mobile device, and having a history of neck or shoulder pain were significant predictors of neck and shoulder pain severity (p 0.018) (Table 5). However, in the multivariate model, both having a history of neck or shoulder pain (OR: -0.213, 95% CI: 0.639 -0.748, p<0.001) and the hand-side used for writing (OR: -0.072, 95%CI: -1.18 - -0.081, p=0.025) were significant predictors of neck and shoulder pain severity, and they both explained 11.3% of its variance. (Table 6).”

The reviewers’ comments were addressed, and the authors responses are below:

Reviewers' comments:

Reviewer's Responses to Questions

Comments to the Author

1. Is the manuscript technically sound, and do the data support the conclusions?

Reviewer #1: Partly

Reviewer #2: Yes

We thank the reviewer for this comment. The entire manuscript has been edited to improve the quality and to make the study replicable. More information has been added to explain the study sample size and the discussion and conclusion were modified to reflect the results of the study.

2. Has the statistical analysis been performed appropriately and rigorously?

Reviewer #1: I Don't Know

Reviewer #2: Yes

For the statistical analysis of this study, we consulted a statistician, and he indicated that the statistical analysis used in this study is appropriate and rigorous. In this study, univariate regression model was used to identify factors that predict the outcomes of interest and to evaluate the proportion that each factor can explain the outcomes of interest. While the multivariate regression model was used to determine the best fit model.

3. Have the authors made all data underlying the findings in their manuscript fully available?

Reviewer #1: No

Reviewer #2: Yes 

The data will be available without restriction

4. Is the manuscript presented in an intelligible fashion and written in standard English?

Reviewer #1: Yes

Reviewer #2: Yes

5. Review Comments to the Author

Please use the space provided to explain your answers to the questions above. You may also include additional comments for the author, including concerns about dual publication, research ethics, or publication ethics. (Please upload your review as an attachment if it exceeds 20,000 characters).

Responses to Reviewer #1: 

Reviewer 1 Authors responses

For lines 91 and 92, please check out the numbers and the percentages. Thank you for the most important point. 

It has been corrected on page 10, lines 8+9, as follows:

“The overall average of hours per day spent on smartphones revealed a maximum percentage of 465 (53.63%) equal to or greater than 4 hours and the lowest percentage of 402 (46.37%) less than 4 hours.”

For lines 96-98, please check out the numbers and the percentages.

 Thank for your kind response 

It has been corrected on page 10, lines 14+15, as follows:

For test massages sent per day, the highest percentage was for those who sent 22 test massages or more 317 (29.99%), followed by 1-5 messages with 260 (29.99%), and the lowest percentages of 12-15 with 32 (3.69%).

Please add the word hour(s) to Table 1 for the variable - average time spent with mobile phones. Thank you for your good comment.

It has been added in to the table 1

Please add the percentages for the variable purpose of use - playing games. I appreciated your comment. 

It has been added on Table 1.

Page 12, for lines 238 and 239, please check out the percentages. Thank for your important comments

It has been corrected on page 15, lines 2- 3, as follows:

In Figure 7, the findings of the current study are shown to be that 292 (33.7%) of the participants had mild pain, 233 (26.9%) had moderate pain, and 89 (10.3%) had severe pain.

Page 13, for lines 249-250, it was written “(OR: -250 0.213, 95% CI: -2.357 to -1.268, p<0.001)” for Table 4. Really?

 It has been corrected on page 16, lines 2- 6, as follows:

In the multivariate model, only having a history of neck or shoulder pain (OR: 0.647, 95%CI: 0.567 - 0.738, p≤0.001) and the number of hours spent on the device for studying (OR.19, 95% CI: 0.254-0.512, p<0.001) were significant predictors of neck and shoulder pain duration in multivariate model, and they both explained 8.4% of its variance (Table 4).

Page 13, for lines 258-260, repetition of the same paragraph which was written above.

It is stated that “(OR: -250 0.213, 95% CI: -2.357 to -1.268, p<0.001)” is written in Table 4 for lines 249-250 on page 13. Really? Thank you very much for your comments.

It has been corrected on page 16, lines 13- 16, as follows:

“in the multivariate model, both having a history of neck or shoulder pain (OR: -0.213, 95% CI: 0.639 -0.748, p<0.001) and the hand-side used for writing (OR: -0.072, 95%CI: -1.18 - -0.081, p=0.025) were significant predictors of neck and shoulder pain severity, and they both explained 11.3% of its variance. (Table 6).”

According to me, in the discussion section, links/comparisons to other literature are poor. The discussion section should be rewritten. Thank you so much for your comments. 

The discussion and conclusion sections have been edited and corrected in red to reflect the findings of the current study.

Responses to Reviewer #2: 

Reviewer 2 Authors responses

Abstract Section

Mention the objectives correctly. Thank you for the most important point. 

The conclusion of the abstract has been corrected on page 2, lines 4-12, as follows:

Objective: Neck and shoulder pain has been linked to prolonged periods of flexed neck posture. However, the influences of factors related to individuals’ characteristics and the time duration and position of using smartphones on the severity and duration of neck and shoulder pain among university students are not well studied. The aim of this study was to identify factors related to individual demographic, history of neck pain, and time duration and positions of using the smart phone that could associate with neck pain severity and duration, and to determine the influence of these factors on neck pain severity and duration among university students.

Conclusion in the abstract section should not include the future scope. Mention clinical implication of the current study in the conclusion

 Thank for your kind response

The conclusion of the abstract has been corrected on page 3, lines 19-23, as follows:

The results of this study may be utilized to pinpoint smartphone usage factors associated with neck and shoulder pain severity and duration. Further, the findings of this study might help to develop preventive strategies to lower the impacts of these factors on the development of neck and shoulder pain severity and duration among university students 

Main Manuscript

Methods

1. Give statistics of how many questionnaires were rolled out and how many filled with Response rate calculation Thank you for your good comment.

It has been edited and added on page 6, lines 23- 24, as follows:

The questionnaires were distributed to the five health faculties by the authors. Questionnaires were distributed to students via Facebook, an online social media and social networking service, by posting them in their batch groups. We received a total of 1000 questionnaires, of which 867 were included and 133 were excluded due to missing data.

2. Exclusion criteria to be mentioned

 I appreciated your comment. 

The exclusion criteria have been clarified on page 7, line 23, and page 8, lines 1-2, as follows:

Students were excluded from the study if they had any of the following chronic medical conditions: musculoskeletal disorders or a prior surgery in the neck, shoulder, or upper limb.

3. Process of development of questionnaire and how its validity was done to be mentioned.

 Thank for your important comments

It has been added on page 7, lines 14- 16, as follows:

The English version of the survey was reliable and valid. The English version of the survey was translated into Arabic using back translation for the Arabic sample (Kristine Bandgaard 2019).

4. In inclusion criteria, give operational definition for duration of usage of smartphone. Thank you for your very important comments.

It has been defined on page 7, line 20+ page 8, lines 1- 2 as your requested.

Discussion to be written in detail with reason for all the factors to have an impact on neck pain. Thank you again and again for the very important comments. The discussion was rewriting as you requested

6. PLOS authors have the option to publish the peer review history of their article (what does this mean?). If published, this will include your full peer review and any attached files.

Do you want your identity to be public for this peer review? For information about this choice, including consent withdrawal, please see our Privacy Policy.

Reviewer #1: No

Reviewer #2: Yes: Prachita Walankar

---

## [Decision Letter · Decision Letter 1]

10 Apr 2023

PONE-D-22-35345R1Neck pain associated with smartphone usage among university studentsPLOS ONE

Dear Dr. Maayah,

Thank you for submitting your manuscript to PLOS ONE. After careful consideration, we feel that it has merit but does not fully meet PLOS ONE’s publication criteria as it currently stands. Therefore, we invite you to submit a revised version of the manuscript that addresses the points raised during the review process. The manuscript should include accurate statistical analysis.

We look forward to receiving your revised manuscript.

Kind regards,

Sıdıka Bulduk, Prof. Dr.

Academic Editor

PLOS ONE

Additional Editor Comments:

I am very much thankful to the reviewers for their deep and thorough review.

Major revisions are required. Please check the reviewers’ comments carefully.

Reviewers' comments:

Reviewer's Responses to Questions

**Comments to the Author**

1. If the authors have adequately addressed your comments raised in a previous round of review and you feel that this manuscript is now acceptable for publication, you may indicate that here to bypass the “Comments to the Author” section, enter your conflict of interest statement in the “Confidential to Editor” section, and submit your "Accept" recommendation.

Reviewer #1: (No Response)

Reviewer #2: All comments have been addressed

Reviewer #3: (No Response)

2. Is the manuscript technically sound, and do the data support the conclusions?

Reviewer #1: Yes

Reviewer #2: Yes

Reviewer #3: (No Response)

3. Has the statistical analysis been performed appropriately and rigorously? 

Reviewer #1: I Don't Know

Reviewer #2: Yes

Reviewer #3: No

4. Have the authors made all data underlying the findings in their manuscript fully available?

Reviewer #1: Yes

Reviewer #2: Yes

Reviewer #3: (No Response)

5. Is the manuscript presented in an intelligible fashion and written in standard English?

Reviewer #1: Yes

Reviewer #2: Yes

Reviewer #3: Yes

6. Review Comments to the Author

Reviewer #1: Why the sentences are different which explain the aim of the study in the abstract and the introduction section?

Page 12, Line 22, I suggest, 317 (36.56%)

The conclusion section should be clear and concise.

Conclusions: The findings of this study may help academic institutions and collegiate students be aware of the negative impact of using smartphones and identify those measures that can contribute to the development of neck and shoulder pain duration and severity. Furthermore, academic institutions may consider providing students with study materials that require using a smartphone for a short time in an attempt to lower the risk and severity of musculoskeletal pain among the young academic population

Reviewer #2: Yes.The authors have answered to all queries raised during first review. This has enhanced the quality of study.

Reviewer #3: The goal of this manuscript intended to identify factors associated with neck pain severity and duration among university students using smartphone. They conduct a cross-sectional study with online questionnaire and analyzed 867 responses. They reported the significant association of neck and shoulder pain duration or severity with a history of neck or shoulder pain, the hand-side used for writing, and the number of hours spent on the device for studying.

1. Line 11 in page 9. “linear regression model…..OR: odds ratios…” how to present OR in linear regression?

2. Table 1. What’s the total time for “The average time spent using a mobile phone (hour)”? Per day, per week, or others? Same question applies to other parts of the table.

3. Table 1. are “playing games”, “social media”, and “studying” exclusive? i.e. it should be “playing games only” and so on?

4. Tables 3, 4, 5, and 6. What analysis was performed for the results in Tables 3, 4, 5, and 6? The title says “linear regression”. If it is linear regression, what’s OR presented in the table? The same question applies to the corresponding manuscript text as well!

7. PLOS authors have the option to publish the peer review history of their article (what does this mean?). If published, this will include your full peer review and any attached files.

Reviewer #1: No

Reviewer #2: No

Reviewer #3: No

<quillbot-extension-portal></quillbot-extension-portal>

---

## [Author Response · Author response to Decision Letter 1]

19 Apr 2023

Reviewers' comments:

Reviewer's Responses to Questions

Comments to the Author

1. If the authors have adequately addressed your comments raised in a previous round of review and you feel that this manuscript is now acceptable for publication, you may indicate that here to bypass the “Comments to the Author” section, enter your conflict of interest statement in the “Confidential to Editor” section, and submit your "Accept" recommendation.

Reviewer #1: (No Response)

Reviewer #2: All comments have been addressed

Reviewer #3: (No Response)

2. Is the manuscript technically sound, and do the data support the conclusions?

Reviewer #1: Yes

Reviewer #2: Yes

Reviewer #3: (No Response)

3. Has the statistical analysis been performed appropriately and rigorously?

Reviewer #1: I Don't Know

Reviewer #2: Yes

Reviewer #3: No

4. Have the authors made all data underlying the findings in their manuscript fully available?

Reviewer #1: Yes

Reviewer #2: Yes

Reviewer #3: (No Response)

5. Is the manuscript presented in an intelligible fashion and written in standard English?

Reviewer #1: Yes

Reviewer #2: Yes

Reviewer #3: Yes

6. Review Comments to the Author

Reviewer #1: Why the sentences are different which explain the aim of the study in the abstract and the introduction section?

The aim of the study has been corrected and it is being consistent in both the abstract and intro section as the following:

Page 6, lines 7-11, as follows: 

The aim of this study was to identify factors related to individual demographics, the history of neck pain, and the time duration and positions of using the smartphone that could be associated with neck pain severity and duration and to determine the influence of these factors on neck pain severity and duration among university students.

Page 12, Line 22, I suggest, 317 (36.56%) 

Page 10, line 8. 

It has been corrected, as per your request.

The conclusion section should be clear and concise.

Conclusions: The findings of this study may help academic institutions and collegiate students be aware of the negative impact of using smartphones and identify those measures that can contribute to the development of neck and shoulder pain duration and severity. Furthermore, academic institutions may consider providing students with study materials that require using a smartphone for a short time in an attempt to lower the risk and severity of musculoskeletal pain among the young academic population.

These two sentences have been deleted. 

Reviewer #2: Yes. The authors have answered to all queries raised during first review. This has enhanced the quality of study.

Reviewer #3: The goal of this manuscript intended to identify factors associated with neck pain severity and duration among university students using smartphone. They conduct a cross-sectional study with online questionnaire and analyzed 867 responses. They reported the significant association of neck and shoulder pain duration or severity with a history of neck or shoulder pain, the hand-side used for writing, and the number of hours spent on the device for studying.

1. Line 11 in page 9. “linear regression model, OR: odds ratios…” how to present OR in linear regression?

We apologize for the mistake, the odd ratios have been removed from the tables and the manuscript.

2. Table 1. What’s the total time for “The average time spent using a mobile phone (hour)”? Per day, per week, or others? Same question applies to other parts of the table. 

 The questions asked about the number of hours spent on smartphones and the number of texts sent per day. The table shows the number of students stratified by the number of hours, and we did not collect data about the total number of hours.

3. Table 1. are “playing games”, “social media”, and “studying” exclusive? i.e. it should be “playing games only” and so on?

Thank you for your comment, but table 1 represents the questionnaire as is.

4. Tables 3, 4, 5, and 6. What analysis was performed for the results in Tables 3, 4, 5, and 6? The title says “linear regression”. If it is linear regression, what’s OR presented in the table? The same question applies to the corresponding manuscript text as well!

The statistical analysis used was linear regression, and we agree with the reviewer that the Odd ratio is not appropriate to be used in this case and the β coefficients are the regression coefficients that are appropriate to be used to indicate the difference between two marginal means in continuous measures. The odd ratios have been removed from the tables and the manuscript.________________________________________

7. PLOS authors have the option to publish the peer review history of their article (what does this mean?). If published, this will include your full peer review and any attached files.

Do you want your identity to be public for this peer review? For information about this choice, including consent withdrawal, please see our Privacy Policy.

Reviewer #1: No

Reviewer #2: No

Reviewer #3: No

---

## [Decision Letter · Decision Letter 2]

24 Apr 2023

Neck pain associated with smartphone usage among university students

PONE-D-22-35345R2

Dear Dr. Maayah,

We’re pleased to inform you that your manuscript has been judged scientifically suitable for publication and will be formally accepted for publication once it meets all outstanding technical requirements.

Kind regards,

Sıdıka Bulduk, Prof. Dr.

Academic Editor

PLOS ONE

Additional Editor Comments (optional):

Reviewers' comments:

Reviewer's Responses to Questions

**Comments to the Author**

1. If the authors have adequately addressed your comments raised in a previous round of review and you feel that this manuscript is now acceptable for publication, you may indicate that here to bypass the “Comments to the Author” section, enter your conflict of interest statement in the “Confidential to Editor” section, and submit your "Accept" recommendation.

Reviewer #1: (No Response)

Reviewer #3: (No Response)

2. Is the manuscript technically sound, and do the data support the conclusions?

Reviewer #1: Yes

Reviewer #3: (No Response)

3. Has the statistical analysis been performed appropriately and rigorously? 

Reviewer #1: Yes

Reviewer #3: (No Response)

4. Have the authors made all data underlying the findings in their manuscript fully available?

Reviewer #1: Yes

Reviewer #3: (No Response)

5. Is the manuscript presented in an intelligible fashion and written in standard English?

Reviewer #1: Yes

Reviewer #3: (No Response)

6. Review Comments to the Author

Reviewer #1: (No Response)

Reviewer #3: (No Response)

7. PLOS authors have the option to publish the peer review history of their article (what does this mean?). If published, this will include your full peer review and any attached files.

Reviewer #1: No

Reviewer #3: No

<quillbot-extension-portal></quillbot-extension-portal>

---

## [Editor Report · Acceptance letter]

14 Jun 2023

PONE-D-22-35345R2 

Neck pain associated with smartphone usage among university students 

Dear Dr. Maayah:

I'm pleased to inform you that your manuscript has been deemed suitable for publication in PLOS ONE. Congratulations! Your manuscript is now with our production department. 

Kind regards, 

on behalf of

Dr. Sıdıka Bulduk 

Academic Editor

PLOS ONE